

# Factors that contributed to Ontario adults' mental health during the first 16 months of the COVID-19 pandemic: a decision tree analysis

Katie J. Shillington[1,2,3], Leigh M. Vanderloo[4,5], Shauna M. Burke[1,6,7], Victor Ng[8,9], Patricia Tucker[1,5,7] and Jennifer D. Irwin[1,6]

[1] Health and Rehabilitation Sciences, Faculty of Health Sciences, University of Western Ontario, London, Ontario, Canada
[2] Department of Neurobiology, University of California San Diego, San Diego, California, United States
[3] Center for Empathy and Social Justice in Human Health, T. Denny Sanford Institute for Empathy and Compassion, University of California San Diego, San Diego, California, United States
[4] Child Health Evaluative Science, The Hospital for Sick Children, Toronto, Ontario, Canada
[5] School of Occupational Therapy, Faculty of Health Sciences, University of Western Ontario, London, Ontario, Canada
[6] School of Health Studies, Faculty of Health Sciences, University of Western Ontario, London, Ontario, Canada
[7] Children's Health Research Institute, London, Ontario, Canada
[8] Schulich School of Medicine and Dentistry, University of Western Ontario, London, Ontario, Canada
[9] Division of Professional Development and Practice Support, College of Family Physicians of Canada, Mississauga, Ontario, Canada

Corresponding author
Katie J. Shillington, kshilli4@uwo.ca

## ABSTRACT

The COVID-19 pandemic has negatively impacted the mental health of individuals globally. However, less is known about the characteristics that contributed to some people having mental health problems during the pandemic, while others did not. Mental health problems can be understood on a continuum, ranging from acute (*e.g.*, depression following a stressful event) to severe (*e.g.*, chronic conditions that disrupt everyday functioning). Therefore, the purpose of this article was to generate profiles of adults who were more or less at risk for the development of mental health problems, in general, during the first 16-months of the COVID-19 pandemic in Ontario, Canada. Data were collected *via* online surveys at two time points: April-July 2020 and July-August 2021; 2,188 adults ($M_{age}$ = 43.15 years; $SD$ = 8.82) participated. Surveys included a demographic questionnaire and four previously validated tools to measure participants' mental health, subjective wellbeing, physical activity and sedentary behaviour, and sleep. A decision tree was generated at each time point for those with *mental health problems*, and those with *no mental health problems*. Results showed that subjective wellbeing was the biggest contributor to mental health status. Characteristics associated with *no mental health problems* among adults included having good wellbeing, being a good sleeper (quantity, quality, and patterns of sleep), and being over the age of 42. Characteristics associated with *mental health problems* included having poor wellbeing and being a poor sleeper. Findings revealed that specific characteristics interacted to contribute to

adults' mental health status during the first 16 months of the COVID-19 pandemic. Given that wellbeing was the biggest contributor to mental health, researchers should focus on targeting adults' wellbeing to improve their mental health during future health crises.

# INTRODUCTION

Mental health is an integral component of overall health and has been conceptualized by the *World Health Organization (2022a)* as a state of mind that enables people to recognize their potential, cope with the demands of life, learn and work well, and contribute to their communities. Mental health is not merely the absence of mental illness; rather, it exists on a continuum that consists of the presence and absence of mental illness and mental health symptoms (*Keyes, 2002*). The ways individuals experience mental health along the continuum vary and can fluctuate in response to changing environments, adverse experiences, and life events (*Headey & Wearing, 1989*; *Peter et al., 2021*; *World Health Organization, 2022b*). Individual, social, and structural factors can interact to protect or undermine a person's mental health (*Oliveros, Agulló-Tomás & Márquez-Álvarez, 2022*; *World Health Organization, 2022b*).

Whereas mental health can be characterized by good behavioural adjustment, the ability to cope with the ordinary stresses of life (*American Psychological Association, 2023*), the ability to adapt and self-manage, and participation in valued roles (*e.g.*, family, work; *Manwell et al., 2015*), subjective wellbeing is a distinct construct (*Lereya, Patalay & Deighton, 2022*) and measures a person's *satisfaction* with life as a whole and/or the various domains of life (*Cummins, Lau & Stokes, 2004*; *International Wellbeing Group, 2013*). Subjective wellbeing is a component of quality of life, which can include both objective (*e.g.*, the degree of physical disability) and subjective (*e.g.*, perceived stress) dimensions (*Cummins, Lau & Stokes, 2004*). Though subjective wellbeing encompasses a person's satisfaction with their health, it also includes a variety of non-health-related domains, such as satisfaction with their standard of living, achievement in life, relationships, safety, community-connectedness, and future security (*International Wellbeing Group, 2013*); it is the totality of these constructs that makes up a person's subjective wellbeing (hereafter referred to as wellbeing). It is worth noting that wellbeing has been found to be positively associated with mental health (*Grant, Guille & Sen, 2013*; *Lombardo et al., 2018*; *Watson & Naragon-Gainey, 2010*). Notably, *Lombardo et al. (2018)* assessed the relationship between mental health and life satisfaction (*i.e.*, wellbeing) and concluded that high life satisfaction was strongly associated with positive mental health among individuals living in Canada. Additionally, in a review conducted by *Watson & Naragon-Gainey (2010)* the authors found that wellbeing was significantly negatively correlated with depression. Similarly, *Grant, Guille & Sen (2013)* concluded that individuals who experienced low levels of wellbeing were at increased risk for depressive symptoms. It is thus clear that wellbeing

predicts mental health status, such as future depression risk (*Grant, Guille & Sen, 2013*); however, it is possible for a serious adverse event, such as the COVID-19 pandemic, to depress an individual's mental health and wellbeing (*Cummins, Lau & Stokes, 2004*; *Headey & Wearing, 1989*). Thus, while it is evident that good wellbeing was associated with positive mental health pre-pandemic, it is critical to assess if such intuitive associations persisted in the face of a global pandemic. Notably, given the evolving nature of the COVID-19 pandemic, *Martin-Maria, Lara & Forsman (2023)* emphasized the importance of studying how the COVID-19 pandemic has affected subjective wellbeing and mental health.

Those exposed to significant adverse events and/or negative individual, social, and structural factors are at a heightened risk of experiencing mental health problems (*Canadian Mental Health Association, 2021*) and poor wellbeing (*Kettlewell et al., 2020*). Mental health problems can range from acute, temporary problems such as depression following a stressful event, to severe chronic conditions that disrupt everyday functioning (*Scheid & Brown, 2010*). In 2018, it was estimated that globally, 506 million adults aged 30–69 years experienced a mental health problem (*Institute for Health Metrics & Evaluation, 2023*). Anxiety and depression were the leading mental health problems in both adult men and women, with depression being most prevalent (*Institute for Health Metrics & Evaluation, 2023*). In Canada, the prevalence of mental health problems is concerning as one in five people experience a mental health problem in any given year, and by the age of 40, approximately 50% of the population will experience, or have experienced, a mental health problem (*Canadian Mental Health Association, 2021*). There is a known relationship between mental health problems and chronic physical conditions (*e.g.*, *Canadian Mental Health Association, 2008*; *Evans et al., 2005*; *Patten, 1999*; *Stein et al., 2019*), and mental health problems have been identified to negatively impact an individual's social (*Kupferberg, Bicks & Hasler, 2016*) and cognitive functioning (*Bunce et al., 2008*; *De Pue et al., 2021*) and contribute to the adoption of poor health behaviours (*e.g.*, *Canadian Mental Health Association, 2008*; *Hoang, Kristoffersen & Li, 2019*; *Parletta, Aljeesh & Baune, 2016*).

Public health emergencies, such as the COVID-19 pandemic, can exacerbate pre-existing mental health problems and introduce new challenges, both of which can have long-term impacts (*Moreno et al., 2020*; *World Health Organization, 2022b*). For example, many individuals fear infection, long-COVID, and death due to COVID-19, either for themselves or their loved ones (*Quadros et al., 2021*; *Hossain et al., 2023*). As a way to mitigate the spread of the virus, in earlier months of the pandemic, public health measures were implemented globally, including national and localized quarantines, lockdowns, restrictions of mass gatherings, physical distancing measures, compulsory mask wearing, and school closures (*Ayouni et al., 2021*). Though the measures were implemented to protect peoples' health at a population level, they inadvertently contributed to negative mental health experiences including social isolation, loneliness (*McQuaid et al., 2021*; *Su et al., 2023*), feelings of helplessness (*Polizzi, Lynn & Perry, 2020*), and strained relationships (*Ellyatt, 2022*). The COVID-19 pandemic also exacerbated global pre-existing inequities, including poverty and unemployment rates

(*Warren & Bordoloi, 2020*), both of which are known risk factors for mental health problems (*Weich & Lewis, 1998*). Additionally, due to the rapid spread of information, the COVID-19 "infodemic" resulted in the dissemination of false information, including rumors and intentional disinformation, which has undermined individuals' mental and physical health, and wellbeing (*Elbarazi et al., 2022*; *Rachul et al., 2020*; *World Health Organization, 2020*). Such stressors have led to reported mental health problems at a global level, including psychological distress, depression, anxiety, post-traumatic stress disorder (*World Health Organization, 2022b*), and poor wellbeing (*Elbarazi et al., 2022*). Specifically, during the first year of the pandemic, researchers estimated that the prevalence of depression and anxiety increased globally by 28% and 26%, respectively (*Santomauro et al., 2021*).

While depression and anxiety have increased on a global level, there have been substantial implications at the national level as well. For instance, according to *Statistics Canada (2021)*, one in four adults living in Canada reported symptoms of depression, anxiety, or post-traumatic stress disorder in Spring 2021, compared to the Fall of 2020 wherein the prevalence was one in five. Specifically, the proportion of adults aged 25–44 years who reported symptoms of depression and/or anxiety increased from 18% to 23%, and 15% to 20%, respectively (*Statistics Canada, 2021*). Further, a greater number of Canadian adults (aged 25–64 years) reported symptoms for at least one mental health disorder from Fall 2020 to Spring 2021 (*Statistics Canada, 2021*). Of the individuals living in Canada who reported symptoms for at least one mental health disorder, 94% reported experiencing negative impacts related to the COVID-19 pandemic, including loneliness, physical health conditions, and/or relationship challenges (*Statistics Canada, 2021*). It is important to note that reporting symptoms for a mental disorder does not always indicate the presence of a mental disorder, as the self-report instruments used in the Statistics Canada Survey on COVID-19 and Mental Health measured the prevalence of mental health disorder *symptoms* and *probable diagnoses* (*Statistics Canada, 2021*).

In addition to the probable pandemic-related mental health disorder diagnoses described above, *Jenkins et al. (2021)* concluded that adults in Canada ($n = 3,000$) reported a deterioration in mental health during the initial stages of the COVID-19 pandemic (*i.e.*, May 2020). This aligns with findings from *Dozois & Mental Health Research Canada (2021)*, who explored how adults in Canada ($n = 1,803$) were coping with the COVID-19 pandemic and its impact on experiences of anxiety and depression. While the author did not specify the timeframe during which data were collected, they concluded that since the onset of the COVID-19 pandemic, the number of participants with anxiety and depression increased from 5% to 20% and 4% to 10%, respectively (*Dozois & Mental Health Research Canada, 2021*). Additionally, *Capaldi, Liu & Dopko (2021)* compared adults' self-reported mental health from pre-pandemic (January–December 2019) to the second wave of the pandemic in Canada (September–December 2020). Despite the fact that over half of the participants reported positive mental health, the authors found that, compared to pre-pandemic, there were significantly fewer participants who reported high levels of positive mental health during the second wave of the pandemic (*Capaldi, Liu & Dopko, 2021*). Building off these findings, *Capaldi et al. (2022)* investigated whether self-reported positive

mental health and perceived change in mental health differed from the second (September–December 2020) to the third (February–May 2021) wave of the COVID-19 pandemic in Canada. The authors found that adults' mental health deteriorated from the second to the third wave of the pandemic, with fewer participants reporting positive mental health and improved mental health at the latter time point (*Capaldi et al., 2022*).

It is clear, based on the above review of literature, that the COVID-19 pandemic has negatively impacted the mental health of adults globally; however, less is known regarding the characteristics that contributed to some people having mental health problems during the first 16 months of the COVID-19 pandemic in Ontario, Canada, while others did not. Understanding this is critical, as the first 16 months of the pandemic included ongoing adaptations and restrictions in attempting to mitigate the severity of health issues (*Canadian Institute for Health Information, 2022*). Of necessity, most of the foci during this period were on developing vaccinations and advocating for preventive measures like physical distancing (*Canadian Institute for Health Information, 2022*; *Chu et al., 2020*; *Government of Canada, 2020a*, *2022b*). Of the provinces in Canada, Ontario was among few with strict preventative measures in place during the early months of the pandemic (*Dekker & Macdonald, 2022*). Thus, understanding the characteristics that contributed to some adults' having mental health problems during this timeframe, while others did not, is instructive to inform the development of programs/interventions aimed at mitigating the negative impact of the COVID-19 pandemic on adults' mental health, as well as creating tailored interventions, responsive to the characteristics of those who had the highest incidence of mental health problems. As such, the purpose of this study was to generate profiles of Ontario adults who were more or less at risk for the development of mental health problems during the first 16-months of the COVID-19 pandemic.

## METHODS

### Study design and participants

This study represents part of an ongoing, longitudinal, survey-based research project titled *Health Outcomes for adults during and following the COVID-19 PandEmic (HOPE)*, which was designed to assess adults' lifestyle-related health behaviours and outcomes, including physical activity, sedentary behaviour, sleep, diet, mental health, wellbeing, and prosocial behaviour, during and following the COVID-19 pandemic in Ontario, Canada (*Shillington et al., 2021*, *2022a*, *2022b*, *2023a*, *2023b*). The current paper includes data collected at time point 1 (April 24–July 13, 2020) and time point 3 (July 29–August 30, 2021) to generate profiles of Ontario adults who were at more or less risk for the development of mental health problems during the first 16-months of the COVID-19 pandemic. The study was approved by the Health Sciences Research Ethics Board (#115827) at Western University and portions of this text were previously published as part of a thesis (https://ir.lib.uwo.ca/etd/9382/). A more complete description of the methods (*i.e.*, study design, study procedures, recruitment, measures, data analysis) for this research is detailed elsewhere (*Shillington et al., 2021*, *2022a*, *2022b*).

Participants were recruited for the *HOPE* study *via* advertisements on social media platforms, as well as community health centres/medical clinics in Ontario, Canada. Study

participants had to be: (1) an Ontario resident; (2) between the ages of 30–59 years at baseline, as individuals within this age range are at highest risk for losing years of healthy life due to chronic disease (*World Health Organization, 2005*); and (3) able to read and write in English. Interested individuals were asked to click the link in the study advertisement, which directed them to an online survey administered *via* Qualtrics, that included the letter of information, eligibility questions, consent, and the time 1 questionnaires. To provide electronic consent, participants were asked to click "I consent to begin the study", acknowledging that they understood the terms and conditions of participating in the study and were making an informed decision to participate. Data collection for the larger project occurred at three time points: (1) baseline/time 1 (April 24–July 13, 2020); (2) time 2 (July 29–August 30, 2020); and time 3 (July 29–August 30, 2021). Data from time point 2 were not included in the current analysis as it occurred very close to time point 1. As such, it would be unlikely to see differences in decision tree modelling profiles.

## Measures

### Exposures

There were 11 explanatory variables explored in the analysis. These included demographic variables (*n* = 6; gender, age, ethnicity, employment, marital status, and education) which were collected at time 1, and health behaviours/outcomes (*n* = 5; sleep, wellbeing, physical activity, sedentary time, and screen time) which were collected at times 1 and 3. Health behaviours/outcomes were assessed using previously validated scales including the Pittsburgh Sleep Quality Index (PSQI; *Buysse et al., 1988*), the Personal Wellbeing Index-Adult (PWI-A; *International Wellbeing Group, 2013*), and the Global Physical Activity Questionnaire (GPAQ; *Bull, Maslin & Armstrong, 2009*), described below.

The PSQI has been previously validated (Cronbach α = 0.83) and is used to measure adults' quantity, quality, and patterns of sleep on seven domains: (1) subjective sleep quality; (2) sleep latency; (3) sleep duration; (4) habitual sleep efficiency; (5) sleep disturbances; (6) use of sleep medication; and (7) daytime dysfunction (*Buysse et al., 1988*). The seven domains are then summed to yield a total score ranging from 0–21, wherein a score greater than five classifies participants as "poor sleepers" (*Buysse et al., 1988*). The total score was used in the current study, where participants were coded as 0 if they scored between 0 and 4 (indicating "good sleepers") or 1 if they scored between 5 and 21 (indicating "poor sleepers").

The PWI-A has been previously validated (Cronbach's α = 0.70–0.85) and is used to measure subjective wellbeing (*International Wellbeing Group, 2013*). The scale includes seven items that correspond to quality of life domains including: (1) standard of living; (2) health; (3) achievement in life; (4) relationships; (5) safety; (6) community-connectedness; and (7) future security (*International Wellbeing Group, 2013*). It also includes two additional (optional) items: (1) satisfaction with life as a whole; and (2) spirituality/religion (*International Wellbeing Group, 2013*). Data can be interpreted at the individual (domain) level; alternatively, the domains can be summed to yield a total wellbeing score (excluding satisfaction with life as a whole, per the scoring protocol;

*International Wellbeing Group, 2013*). For the purpose of this study, the domains were summed, wherein a score less than 70 indicated poor wellbeing (*International Wellbeing Group, 2013*). As such, participants were coded as 0 if they scored between 0 and 69 (indicating poor wellbeing) or 1 if they scored between 70–100 (indicating good wellbeing). The cut-scores used for this analysis were informed by the authors of the scale, who stated that the normative range for means of Western samples was between 70 and 80 points (*International Wellbeing Group, 2013*). Further, *Tomyn, Weinberg & Cummins (2015)* stated that wellbeing scores at or above 70 points reflect 'normal' levels of wellbeing, while lower scores reflect compromised levels of wellbeing.

The GPAQ has been previously validated (*Bull, Maslin & Armstrong, 2009*) and is used to measure physical activity at the population level. This scale includes four domains: (1) activity at work; (2) travel to and from places; (3) recreational activities; and (4) sedentary behaviour (*Bull, Maslin & Armstrong, 2009*). For the purpose of this study, the domains of recreational activities and sedentary behaviour were used. To yield a total score for recreational-related physical activity (moderate to vigorous physical activity; MVPA), data from the following questions were used: (1) "In a typical week, on how many days do you do *vigorous-intensity* sports, fitness or recreational (leisure) activities?"; (2) "How much time do you spend doing *vigorous-intensity* sports, fitness or recreational activities on a typical day?"; (3) "In a typical week, on how many days do you do *moderate-intensity* sports, fitness or recreational (leisure) activities?"; and (4) "How much time do you spend doing *moderate-intensity* sports, fitness or recreational activities on a typical day?" The total score for recreational-related physical activity was in minutes per week, wherein participants who engaged in 0–149 min of recreational-related physical activity per week were coded as 0 (not meeting recommended amount of MVPA) and participants who engaged in 150 min or greater of recreational-related physical activity per week were coded as one (meeting recommended MVPA), per Canada's 24-h Movement Guidelines for adults (*Canadian Society for Exercise Physiology, 2020*). With respect to sedentary behaviour, participants were asked "How much time do you usually spend sitting or reclining on a typical day?" Participants who engaged in more than 8 h of sedentary pursuits per day were coded as 0 (not meeting recommended sedentary behaviour guidelines) and participants who engaged in 8 h or less of sedentary pursuits per day were coded as 1 (meeting recommended sedentary behaviour guidelines), again as outlined in Canada's 24-h Movement Guidelines (*Canadian Society for Exercise Physiology, 2020*). Lastly, to measure participants' screen use, the following question was used: "How much time do you usually spend watching TV or using a computer, tablet or smartphone on a typical day?" This item was not manually coded, as it was a continuous variable and thus a standard median split was used (calculated *via* SPSS version 29.0). Supplemental 1 outlines the response scale for each variable as well as variable type (*e.g.*, nominal and ordinal) and number of levels.

### Outcome

The outcome variable in the current study was mental health, which was assessed using the Mental Health Inventory-5 (MHI-5; *Berwick et al., 1991*). The MHI-5 has been previously

validated and measures mental health status using five items: general positive affect (two items); anxiety (one item); depression (one item); and behavioural/emotional control (one item; *Berwick et al., 1991*). The items are then summed to yield a total mental health score. While the authors of the scale did not establish a cut score, researchers have used a cut score of 76 and below to indicate mental health problems (*Kelly et al., 2008*). Specifically, *Kelly et al. (2008)* concluded that "a case of common mental disorder is defined by a score of less than or equal to 76 for the MHI-5" (p. 4). As such, participants were coded as 0 if they scored between 0 and 76 (indicating mental health problems) or 1 if they scored between 77–100 (indicating no mental health problems). This said, it is important to recognize that mental health exists on a continuum and should not be interpreted as absolute, as the group names might suggest. While participants who scored between 0 and 76 were included in the same group (*i.e.*, mental health problems), the group represents a range of scores and consequently, varied experiences of mental health problems. Thus, the group represents differed experiences of mental health, ranging from minor to major mental health problems.

## Statistical analyses

Decision tree modeling has been applied in the fields of medicine and public health (*Batterham, Christensen & Mackinnon, 2009*; *Camp & Slattery, 2002*; *Jung et al., 2015*) and can be used to understand features and extract patterns in large datasets (*Myles et al., 2004*). It is a commonly used method for "establishing classification systems based on multiple covariates or for developing prediction algorithms for a target variable" (*Song & Lu, 2015*, p.130). Profiles in the current study (*mental health problems vs. no mental health problems*) were generated based on two broad categories of variables: (1) demographics (gender, age, ethnicity, employment, marital status, and education); and (2) health behaviours/outcomes (sleep, wellbeing, physical activity, sedentary time, and screen time). Some of the health behaviour/outcomes were dichotomized using cut-scores (noted above) and to this end, we relied on a set of binary rules to calculate a target value related to our study objective. No *a priori* hypotheses were put forward as decision tree modeling is a data-driven analysis and requires no formal theoretical structure (*Guerrero et al., 2020*).

Decision tree models for the current study were generated using the exhaustive chi-square automatic interaction detector (CHAID) algorithm (*McArdle & Ritschard, 2014*). Exhaustive CHAID is commonly used when examining large datasets and involves splitting the data into "mutually exclusive, exhaustive, subsets that best describe the dependent variable" (*Kass, 1980*, p. 199). This method has a number of strengths including: (1) it is non-parametric; (2) it is robust against issues regarding missing data and outliers; and (3) all types of variables (continuous, ordinal, categorical) can be included (*Merkle & Shaffer, 2011*). The exhaustive CHAID method begins with a root (or "parent") node that splits into two or more mutually exclusive subsets ("child nodes"; *Song & Lu, 2015*). Nodes continue to split until pre-determined homogeneity or stopping criteria are met (*Song & Lu, 2015*). The following statistical model specifications and stopping criteria were applied in the current study: (1) the significance level for splitting nodes was set at $p < 0.05$; (2) the Bonferroni method was used to obtain the significant values of adjustment; (3) the

minimum change in expected cell frequencies was 0.001; (4) Pearson's $X^2$ was used; (5) model depth was set at 3; (6) the minimum number of cases in parent nodes was set at 100 and in child nodes was set at 50; (7) cross-validation (10-folds) was used to assess the tree structure; and, (8) the mis-classification risk was calculated as a measure of model reliability. Missing values were handled using multiple imputation and data were analyzed in SPSS. Missingness for each covariate was under 15% at time point 1 and 65% at time point 3. Multiple imputation ($N$ = 2,188) was performed in SPSS (version 29) using the fully conditional specification approach to account for bias introduced from missing data (*Van Buuren & Groothuis-Oudshoorn, 2011*). All variables were included in the model and the imputation method was set to automatic. All $p$-values were two-tailed and statistical significance was set at $\alpha$ = 0.05. For likelihood ratio tests for interaction, a threshold of $p < 0.3$ was used (*Harrell, 2015*). Two decision tree models were generated, one utilizing time point 1 data and a second utilizing time point 3 data. All participants were included in the time point 1 decision tree model ($N$ = 2,188); one participant was excluded from the time point 3 decision tree model due to invalid/spurious data.

## RESULTS

### Demographics

A total of 2,188 individuals participated in the study. The mean age of participants was 43.15 years ($SD$ = 8.82), with the majority identifying as female ($n$ = 1,743; 89.55%), of European ancestry ($n$ = 1,789; 91.55%), and having a university degree or higher ($n$ = 1,123; 57.21%). Additionally, most participants in the study were married, common law, or engaged ($n$ = 1,535; 78.20%). Full demographic details have been published elsewhere (*Shillington et al., 2023a*).

### Time point 1 (April 24–July 13, 2020) decision tree

Figure 1 shows the final 3-level model at time 1, comprising 13 nodes, seven of which were terminal subgroups (*i.e.*, nodes that do not split any further). Four predictor variables reached significance (wellbeing, sleep, physical activity, age) and were selected because they best differentiated adults who had mental health problems (79.6%) from those who did not (20.4%) based on the cut scores outlined above. The first level of the tree was split into two initial branches according to participants' perceived wellbeing, meaning that this variable was the best predictor of mental health status (*mental health problems* or *no mental health problems*). The *no mental health problems* group included participants who had good wellbeing (Node 2), were classified as good sleepers (Node 6), and who were over the age of 42 (Node 12; 72.0% *no mental health problems*). The probability decreased when participants were 42 years of age or younger (Node 11; 45.3% *no mental health problems*). Those in the *mental health problems* group had poor wellbeing (Node 1) and were classified as poor sleepers (Node 3; 93.2% *mental health problems*). The probability decreased when participants were classified as good sleepers (Node 4) and were not meeting the MVPA recommendations (Node 8; 87.0% *mental health problems*), and when they were classified as good sleepers (Node 4) and were meeting the MVPA recommendations (Node 7; 75.6% *mental health problems*). Decision rules for the

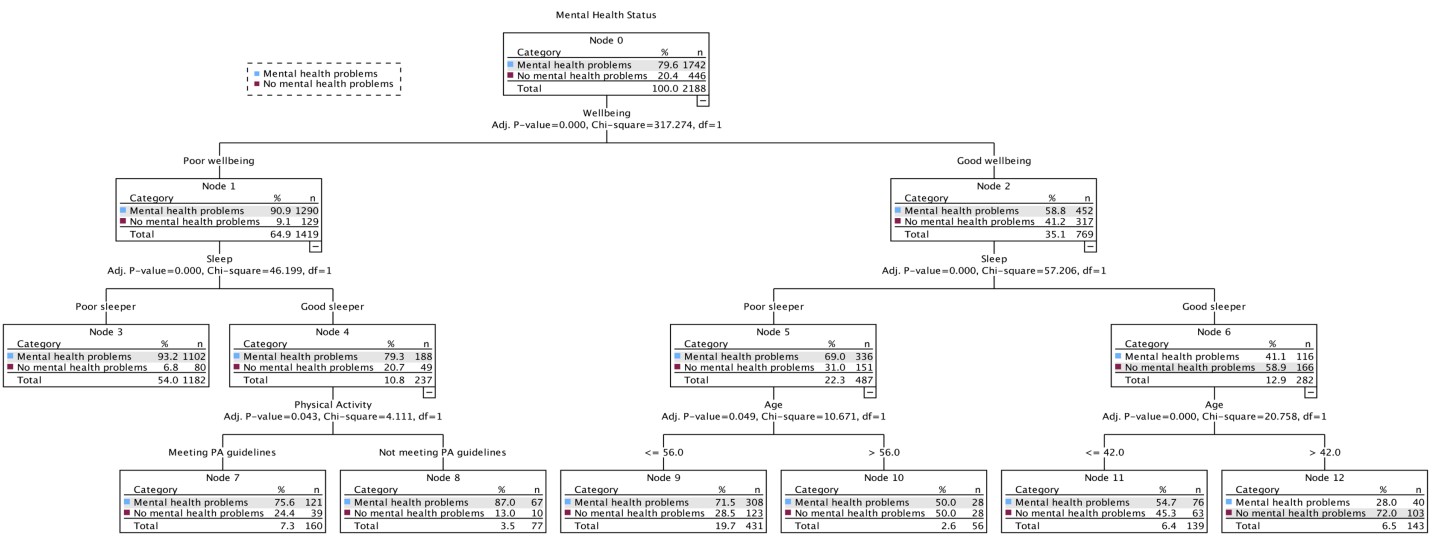

**Figure 1 The classification tree of mental health status using the exhaustive $X^2$ automatic interaction detector (CHAID) method at time point 1 (April 24–July 13, 2020).**

**Table 1 Percentage of classification of mental health problems for terminal nodes at time point 1 (April 24–July 13, 2020), by risk probability based on decision rules using the exhaustive CHAID method.**

| Classification | Node | IF | THEN |
|---|---|---|---|
| 1st | 3 | Participants' wellbeing was *poor* AND they were classified as a *poor* sleeper | 93.2% |
| 2nd | 7 | Participants' wellbeing was *poor* AND they were classified as a *good* sleeper AND they *met* physical activity (MVPA) guidelines | 75.6% |
| 3rd | 8 | Participants' wellbeing was *poor* AND they were classified as a *good* sleeper AND they *did not meet* physical activity (MVPA) guidelines | 87.0% |
| 4th | 9 | Participants' wellbeing was *good* AND they were classified as a *poor* sleeper AND they were *less than or equal to* 56 years of age | 71.5% |
| 5th | 10 | Participants' wellbeing was *good* AND they were classified as a *poor* sleeper AND they were *greater than* 56 years of age | 50.0% |
| 6th | 11 | Participants' wellbeing was *good* AND they were classified as a *good* sleeper AND they were *less than or equal to* 42 years of age | 54.7% |
| 7th | 12 | Participants' wellbeing was *good* AND they were classified as a *good* sleeper AND they were *greater than* 42 years of age | 28.0% |

Note:
Decision rules displayed in plain text. An example of a lay interpretation is as follows: for the 7th classification/Node 12, IF participants' wellbeing was good AND they were classified as a good sleeper AND they were over the age of 42, THEN the probability of them experiencing mental health problems was 28.0%.

prediction of mental health problems at time point 1 are presented in Table 1, which also shows detailed "IF–THEN" rules.

## Time point 3 (July 29–August 30, 2021) decision tree

Figure 2 shows the final 3-level model at time three, comprising eight nodes, five of which were terminal subgroups. Three predictor variables reached significance (wellbeing, sleep, age) and were selected because they best differentiated adults who self-reported mental health problems (76.1%) from those who did not (23.9%). The first level of the tree was split into two initial branches according to participants' perceived wellbeing, meaning that this variable was the best predictor of mental health status (*mental health problems* or *no mental health problems*). The *no mental health problems* group included participants who

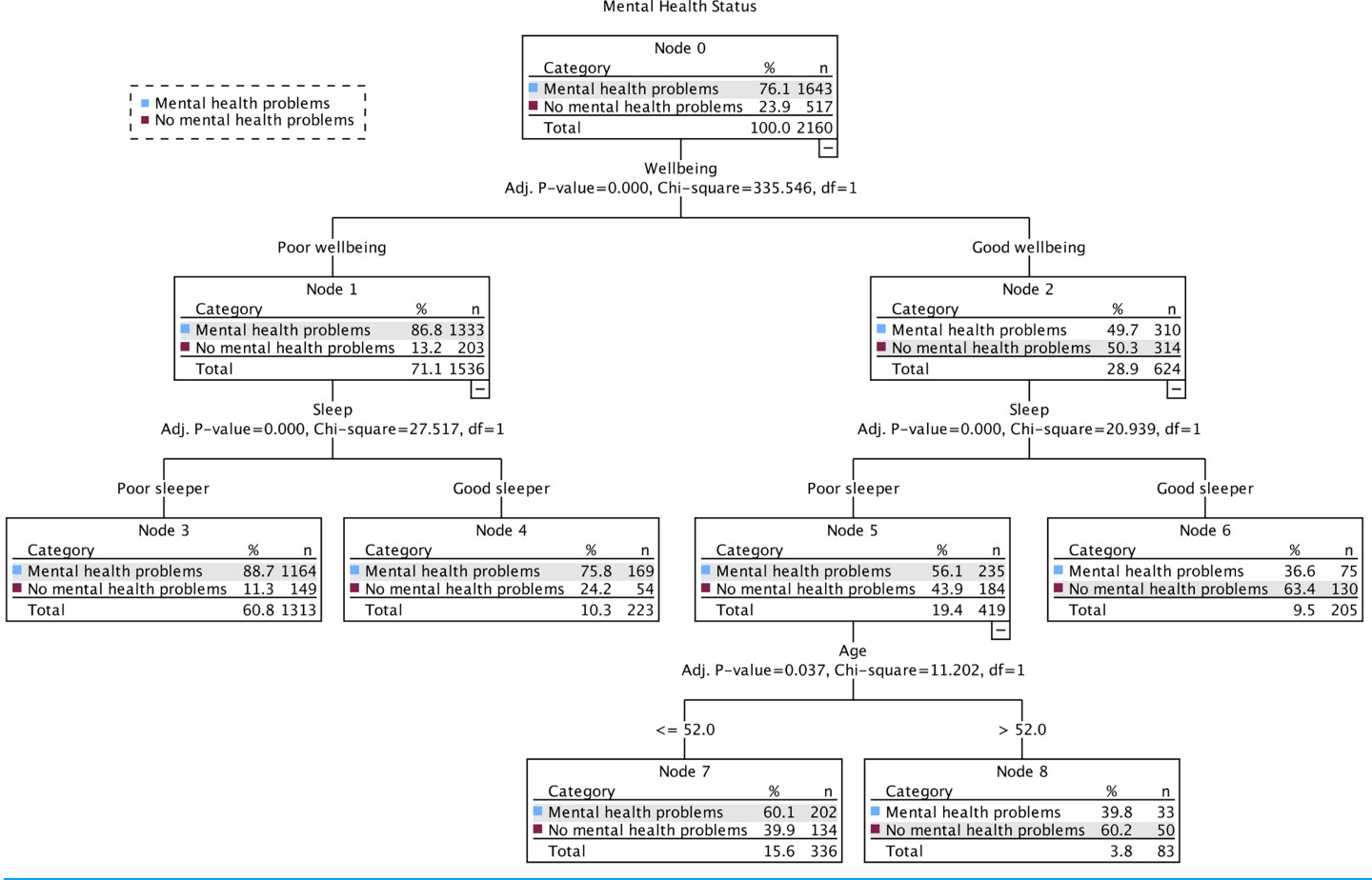

**Figure 2 The classification tree of mental health status using the exhaustive $X^2$ automatic interaction detector (CHAID) method at time point 3 (July 29–August 30, 2021).**

**Table 2 Percentage of classification of mental health problems for terminal nodes at time point 3 (July 29–August 30, 2021), by risk probability based on decision rules using the exhaustive CHAID method.**

| Classification | Node | IF | THEN |
|---|---|---|---|
| 1st | 3 | Participants' wellbeing was *poor* AND they were classified as a *poor* sleeper | 88.7% |
| 2nd | 4 | Participants' wellbeing was *poor* AND they were classified as a *good* sleeper | 75.8% |
| 3rd | 6 | Participants' wellbeing was *good* AND they were classified as a *good* sleeper | 36.6% |
| 4th | 7 | Participants' wellbeing was *good* AND they were classified as a *poor* sleeper AND were *less than or equal to* 52 years of age | 60.1% |
| 5th | 8 | Participants' wellbeing was *good* AND they were classified as a *poor* sleeper AND were *greater than* 52 years of age | 39.8% |

**Note:**
Decision rules displayed in plain text.

had good wellbeing (Node 2) and were classified as good sleepers (Node 6; 63.4% *no mental health problems*). The probability decreased when participants were classified as poor sleepers (Node 5) and were over the age of 52 (Node 8; 60.2% *no mental health problems*) and when they were classified as poor sleepers (Node 5) and were 52 years of age or younger (Node 7; 39.9% *no mental health problems*). Participants in the *mental health*

 

*problems* group included participants who had poor wellbeing (Node 1) and were classified as poor sleepers (Node 3; 88.7% *mental health problems*). The probability decreased when participants were classified as good sleepers (Node 4; 75.8% *mental health problems*). Decision rules for the prediction of mental health problems at time point 3 are presented in Table 2.

## DISCUSSION

The purpose of this article was to use decision tree modeling to generate profiles of adults who were more or less at risk for the development of mental health problems during the first 16-months of the COVID-19 pandemic in Ontario, Canada. The models yielded profiles based on demographic characteristics and health behaviours/outcomes. From April 24–July 13, 2020 (time point 1), characteristics of those in the '*no mental health problems*' group included having good wellbeing, being a good sleeper, and being over the age of 42, while characteristics of those in the '*mental health problems*' group included having poor wellbeing and being a poor sleeper. From July 29–August 30, 2021 (time point 3), characteristics associated with no mental health problems included having good wellbeing and being a good sleeper, while characteristics associated with mental health problems included having poor wellbeing and being a poor sleeper.

Wellbeing was the strongest contributor to mental health status during the first 16 months of the COVID-19 pandemic. In both models (time points 1 and 3), good wellbeing was associated with better mental health (*i.e.*, *no mental health problems*). This finding is not surprising as wellbeing has been positively linked to mental health in previously published work (*Lombardo et al., 2018*). Specifically, data collected pre-pandemic (2003–2012) *via* the Canadian Community Health Survey (CCHS) revealed that life satisfaction (*i.e.*, subjective wellbeing) was strongly associated with self-reported mental health and that individuals ($n = 646,471$) with poor mental health experienced low life satisfaction (*Lombardo et al., 2018*). This aligns with findings from the current study, as participants with poor wellbeing (or quality of life per *International Wellbeing Group, 2013*) were more likely to experience mental health problems at both time points. This might be explained, in part, by the theory of Subjective Wellbeing Homeostasis, which suggests that a person's wellbeing is relatively stable and that while it can change in the short-term, over time an individual's wellbeing will return to a baseline level (*Cummins, 2010*; *Headey & Wearing, 1989*). However, it is possible for an adverse event (*e.g.*, the COVID-19 pandemic) to negatively impact an individual's wellbeing, resulting in their baseline level of wellbeing falling below its normal homeostatic range (*Cummins, Lau & Stokes, 2004*; *Headey & Wearing, 1989*). This was evidenced by *Zajacova et al. (2020)*, who investigated changes in adults' ($n = 4,627$) psychological wellbeing during the early months of the pandemic in Canada. The authors concluded that individuals' psychological wellbeing decreased significantly from March–May 2020 (*Zajacova et al., 2020*). Therefore, it is possible that the first 16 months of the pandemic had a negative toll on participants' wellbeing, potentially contributing to mental health problems.

Experiencing no mental health problems was highest among those who had good wellbeing and sleep (time points 1 and 3), and who were over the age of 42 (time point 3

only), while presence of mental health problems were highest among those who had poor wellbeing and sleep at both time points. These findings are unsurprising as it has been established that good sleep quality is a protective factor against mental health conditions (*Scott et al., 2021*). Specifically, achieving high quality sleep has been found to reduce risk of depression (*Raboch et al., 2017*) and lead to improvements in wellbeing (*Pilcher, Ginter & Sadowsky, 1997*; *Prendergast, Mackay & Schofield, 2016*). In the same vein, poor sleep quality has been associated with mental and physical distress, anxiety, depression (*Strine & Chapman, 2005*), and poor quality of life (*Baldwin et al., 2001*). Since the onset of the COVID-19 pandemic, similar findings have been reported globally. Specifically, *Simonetti et al. (2021)* conducted a study in Morocco during the early stages of the pandemic (February-April, 2020) and found a positive correlation between anxiety and poor sleep quality among adult nurses ($n = 1,005$). Similarly, *Stanton et al. (2020)* conducted a study among Australian adults ($n = 1,491$) and concluded that negative changes in sleep during the early months of the pandemic (April 2020) were associated with higher levels of depression, anxiety, and stress symptoms. In a study of Norwegian adults ($n = 1,281$), *Ernstsen & Havnen (2021)* found that anxiety and depressive symptoms were associated with sleep disturbance during the early stages of the pandemic (June 2020). Moreover, *Hyun et al. (2021)* investigated the impact of mental health symptoms on sleep quality of young adults ($n = 908$; aged 18-30) in the United States during the early pandemic (April-May 2020). The authors concluded that young adults experienced high rates of sleep problems and that anxiety and depressive symptoms were predictors of sleep quality (regardless of pre-existing diagnoses; *Hyun et al., 2021*). This finding aligns with the time point 3 model in the current study, as participants who had good wellbeing and sleep, and who were over the age of 42 did not experience mental health problems. While it is interesting that older age was associated with no mental health problems in the current study, it is not necessarily surprising as data from the 2015–2017 CCHS revealed that older age was associated with positive mental health (*Varin et al., 2020*). Additionally, *Trollor et al. (2007)* concluded that age was a significant predictor of mental health problems among their sample of Australian adults ($n = 10,641$), with increasing age being associated with a lower likelihood of experiencing symptoms of anxiety or affective disorders.

While the same predictor variables reached significance in both models (wellbeing, sleep, age), during the early months of the pandemic (time point 1 only), physical activity (MVPA) appeared to play a significant role with regard to mental health status. Notably, participants who had poor wellbeing, but achieved good sleep and met the Canadian physical activity (MVPA) guidelines were less likely to experience mental health problems (24.4% *no mental health problems*) compared to those who had poor wellbeing, good sleep, and did not meet the Canadian physical activity guidelines (13.0% *no mental health problems*). This finding underscores the importance of physical activity in promoting positive mental health; a finding that has been widely reported (*e.g.*, *Creese et al., 2021*; *Jacob et al., 2020*; *Rebar et al., 2015*). However, it is worth noting that the percentage of participants who experienced no mental health problems in this profile was low compared to those who experienced mental health problems, which suggests that participants' experiences of poor wellbeing might have negatively impacted their ability to engage in

physical activity during the early months of the pandemic. This interpretation aligns with findings from a systematic review conducted by *Violant-Holz et al. (2020)*, where the authors investigated whether physical activity was used as a coping strategy early in the pandemic (January 2019–July 2020). The authors concluded that there was an association between mental health distress and physical activity status, such that the pandemic and related public health protections were associated with psychological distress, which might have hindered individuals' engagement in physical activity (*Violant-Holz et al., 2020*). Thus, while participants who met the physical activity guidelines experienced fewer mental health problems compared to those who did not meet the physical activity guidelines, wellbeing was a stronger contributor to mental health status than physical activity. It is therefore possible that those who experienced poor wellbeing had a difficult time engaging in physical activity as a result. However, it is also worth noting that there may have been other factors, aside from poor wellbeing, that contributed to participants lack of engagement in physical activity, including the closure of gyms and recreational facilities and demographic factors such as living alone, having low income, and loss of employment due to COVID-19 (*Fearnbach et al., 2021*), to name a few.

## Limitations

This study is not without limitations. First, while multiple imputation is a statistically sound method to handle missing data, especially in large datasets (*Clark & Altman, 2003*; *Lee & Huber, 2021*), it is not without limitations. In the current study, data was assumed to be missing at random (MAR), which can lead to biased results in complete case analyses (*Sterne et al., 2009*). To overcome these biases, multiple imputation can be used (*Sterne et al., 2009*), to statistically facilitate the prediction of missing values based on participants with complete data (*Jakobsen et al., 2017*). Further, while percent missingness at time point 1 was less than 15%, the proportion of data missing was much larger at time point 3 (less than 65%). In this case, it is possible that results from the current study might be biased; however, *Lee & Huber (2021)* assessed the effectiveness of multiple imputation in a large dataset ($N = >3,000$) with 20–80% missing data and concluded that data MAR produced reliably accurate estimates even with a large proportion of missing data. Second, the study lacks generalizability because the sample was primarily comprised of White, female-identifying individuals of high socioeconomic status. This is not surprising as participants were primarily recruited through social media platforms, which women reportedly use more than men (*Bush, 2024*). Thus, study findings may be more relevant among these sub-populations. It is also worth noting that the sample make up likely influenced the results, as no demographic characteristics (aside from age, which was a continuous variable) were included in the decision tree models. It is possible that this was due to the disproportionate sample sizes for each group/category. As such, while gender, ethnicity, employment, marital status, and education were not strong predictors of mental health status in the current sample, this is not to say that such variables do not contribute to an individual's mental health status in general. Given that the COVID-19 pandemic has negatively affected minoritized individuals and those of low socioeconomic status, findings from the current study are not representative of such voices and experiences. In the future,
researchers might consider stratifying their sample by targeting groups of diverse ethnic origins, genders, and socioeconomic status in order to achieve greater diversity and representation among the population. Third, the percentage of participants classified as having mental health problems was high at both time points (79.6% at time point 1; 76.1% at time point 2) and might not represent the mental health of the general population. It is possible that the high proportion of participants in this classification was due to the cut-score used to indicate mental health problems, despite it being based on recommendations in the literature (*Kelly et al., 2008*). It is equally plausible that the high percentage of participants classified as having mental health problems is reflective of the timeframe during which data was collected and the nature of the COVID-19 pandemic, though it is important to note that causation cannot be determined. This is crucial to consider when interpreting study findings, as findings might not be generalizable to the broader population. Lastly, it is worth noting that wherever possible, authors relied on pre-established cut-scores (*e.g.*, PSQI, *Buysse et al., 1988*; PWI-A, *International Wellbeing Group, 2013*) to dichotomize the health behaviour/outcome variables for the decision tree models. However, the cut-scores used in the analyses reflected a range of scores, and findings should be interpreted with caution. Notably, to dichotomize wellbeing, participants who scored between 0–69 represented "poor wellbeing", while participants who scored between 70–100 represented "good wellbeing" (*i.e.*, the normative range for means of Western samples). While wellbeing scores below 50 points indicate homeostatic failure (*i.e.*, a drop in wellbeing below the normative range), scores between 51–69 points may represent homeostatic failure or homeostatic normality (*Tomyn, Weinberg & Cummins, 2015*). Thus, it is possible that not all participants included in the "poor wellbeing" group experienced homeostatic failure, based on the cut-score descriptions provided, and that some experienced homeostatic normality of a low set-point. In instances where cut-score thresholds were not available, such was the case with the MHI-5 (*Berwick et al., 1991*), our approach to dichotomizing the outcome variable was informed by recommendations in the literature (*Kelly et al., 2008*). It is worth noting that researchers have used MHI-5 cut-scores ranging from 70 to 76 to indicate mental health problems (*Hoeymans et al., 2004*; *Kelly et al., 2008*; *van den Beukel et al., 2012*).

## Study implications and future directions

Findings from the current study might be particularly useful to program planners and public health personnel alike. Given the toll that the COVID-19 pandemic has had on the mental health of adults living in Canada (*Dozois & Mental Health Research Canada, 2021*; *Jenkins et al., 2021*; *Statistics Canada, 2021*) study findings might aid program planners/ researchers in developing interventions aimed at improving individuals' mental health. The current study revealed that good wellbeing and sleep quality predicted better mental health status. Thus, researchers may consider developing interventions targeting wellbeing and/or sleep to improve individuals' mental health. Additionally, findings from the current study might be relevant to public health personnel during future health crises and could be used to inform health promotion and disease preventative efforts.

## CONCLUSION

Over the past 3 years, individuals around the world have experienced the pervasive impacts of the COVID-19 pandemic; its health repercussions have been devastating, with illness in varying degrees of severity and deaths that continue to plague populations—and concomitantly, their health care systems—worldwide (*Barrett et al., 2020*; *Pollard, Morran & Nestor-Kalinoski, 2020*; *World Health Organization, 2023*). It has been established that the pandemic has negatively impacted the mental health of adults globally (*e.g.*, *McQuaid et al., 2021*; *Statistics Canada, 2021*; *Su et al., 2023*); but an understanding of the characteristics that contributed to mental health problems during the first 16 months of the COVID-19 pandemic, specifically in Ontario, Canada, were lacking. This study revealed that adults with no mental health problems generally had good wellbeing and sleep quality, and were over the age of 42 years, while participants with mental health problems experienced poor wellbeing and had poor sleep quality. Findings from the current study highlight that wellbeing was the biggest contributor to mental health status during the first 16 months of the COVID-19 pandemic. Future interventions/programs should focus on targeting adults' wellbeing in order to improve their mental health during times of health crises.

## ACKNOWLEDGEMENTS

We would like to thank the participants of this study, as well as Dr. Michelle Guerrero for her statistical expertise.

### Funding
The authors received no funding for this work.

### Competing Interests
The authors declare that they have no competing interests.

### Author Contributions
- Katie J. Shillington conceived and designed the experiments, performed the experiments, analyzed the data, prepared figures and/or tables, authored or reviewed drafts of the article, and approved the final draft.
- Leigh M. Vanderloo conceived and designed the experiments, analyzed the data, authored or reviewed drafts of the article, and approved the final draft.
- Shauna M. Burke conceived and designed the experiments, authored or reviewed drafts of the article, and approved the final draft.
- Victor Ng conceived and designed the experiments, authored or reviewed drafts of the article, and approved the final draft.
- Patricia Tucker conceived and designed the experiments, authored or reviewed drafts of the article, and approved the final draft.

- Jennifer D. Irwin conceived and designed the experiments, performed the experiments, analyzed the data, authored or reviewed drafts of the article, and approved the final draft.

### Human Ethics

The following information was supplied relating to ethical approvals (*i.e.*, approving body and any reference numbers):

The Health Sciences Research Ethics Board at the University of Western Ontario granted ethical approval to carry out the study activities.

### Data Availability

Data is available at Open Science Framework: http://doi.org/10.17605/OSF.IO/9WA4V.

### Supplemental Information

Supplemental information for this article can be found online at http://dx.doi.org/10.7717/peerj.17193#supplemental-information.

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
