# Peer review of "Factors that contributed to Ontario adults’ mental health during the first 16 months of the COVID-19 pandemic: a decision tree analysis"

_PeerJ, doi:10.7717/peerj.17193_

## Round 0.1 · original submission · Major Revisions

As you can see, the reviewers have offered constructive feedback that I believe will greatly assist you in revising your manuscript. I kindly request that you provide comprehensive responses to each comment from the reviewers.

Reviewer 1 ·

Basic reporting

The paper is well-written with no major gaps in the literature.

In the abstract specifically, “mental health problems” is too vague, a general description of what this represents is warranted (understanding that is an abstract and therefore needs to be concise).

Figures 1 and 2 are both difficult to read. I recommend recreating these figures outside of the default output from the statistical software to improve readability.

The manuscript is self-contained and included results relevant to the hypotheses but I have some concerns regarding how variables were scored and subsequently analyzed.

Experimental design

Associations between good subjective wellbeing and lack of mental health problems is intuitive, I am not sure this provides helpful information.

The cutoff of 70-100 for good well being and 0-69 appears arbitrary. Please provide additional support for why this is a meaningful cutoff. Intuitively, well-being is a continuous variable, it is difficult to assume that a score of 69 vs 71 is equivalent to 49 vs. 91.

Similarly, the measurement of mental health problems seems arbitrary, I recognize that dichotomizing the variables works for the analysis but practically, more needs to be presented to support the 0/1 dichotomization. Further – describing a score of 77-100 as indicating “no mental health problems” seems to be a stretch. Wouldn’t only a score of 100 = no mental health problems? Minor vs major mental health problems might better reflect the nature of the measure.

The generalizability of the finds needs to be considered. Nearly 90% of respondents were female and 57% had a college degree. The extent that selection factors played a role in who completed the survey, compared to who likely did not complete the survey need to be explored.

Validity of the findings

The findings are likely valid, I just have concerns with the scoring of the measures and the meaningfulness of subject well-being being associated with mental health problems.

Additional comments

I would be curious, and recognize it would change a substantial amount of the paper but I would guess that the findings and outcomes might be a bit different if subjective well-being were not included as a predictor. What are the bivariate correlations among the outcomes and subjective well-being? Other predictors may have more practical importance. Related, the discussion could be strengthened by further describing the clinical and practical importance of the papers - e.g., what are the implications, recommendations for future research or practice?

Reviewer 2 ·

Basic reporting

No comment.

Experimental design

On line 316 the authors mention that they use multiple imputation to handle missing values, however I couldn’t find any further information about how the multiple imputation was done. It would be helpful if the authors would provide additional information on the imputation procedure e.g., assumptions regarding the nature of the missing data, use of imputation strategies like chained equations, which variables were included in the model, and how much data was missing and on which variables.

The authors also mention multiple imputation in the limitations but should elaborate on why this is a potential limitation, e.g., assumptions regarding data being missing at random not met. A complete case analysis could also be a sensitivity analysis – this would be pretty standard when imputation is used in the main analysis.

I noticed that approximately 80% of the sample reports mental health problems at time point 1 and this is similar at time point 3. This is quite a bit different than what would be expected in the general population – it may be worth more explicitly mentioning this and the potential implications of this for the interpretation of the results in the limitations section (I can see it is discussed a bit in lines 438-441 in the context of reverse causation and physical activity).

Validity of the findings

No comment.

Additional comments

This was a well-written article. Beyond the request for some additional information regarding the multiple imputation and consideration of the sample, I think this article meets all other standards.

---

## Round 0.2 · accepted · Accept

Thank you for addressing the reviewers' concerns.

Reviewer 1 ·

Basic reporting

The figures have been improved and the writing is unambiguous and professional. The literature is sufficient.

Experimental design

The clinical cutoffs that were used were appropriate in this study but they do come with limitations - much of which are outside the scope of this paper.

Validity of the findings

I still have some concerns regarding the close association between well-being and mental health outcomes and but the authors have partially mitigated my concerns. I am not convinced that the association between well-being and mental health is that novel but the approach and other associations to add to the literature.

Additional comments

No additional comments